# Temporal Effective Batch Normalization in Spiking Neural Networks

**Chaoteng Duan**[*]
School of Electronic and Computer Engineering
Peking University
Beijing, China 100871
duanchaoteng@stu.pku.edu.cn

**Jianhao Ding**[*]
School of Computer Science
Peking University
Beijing, China 100871
djh01998@stu.pku.edu.cn

**Shiyan Chen**
School of Electronic and Computer Engineering
Peking University
Beijing, China 100871
strerichia002p@stu.pku.edu.cn

**Zhaofei Yu**[†]
Institute for Artificial Intelligence
School of Computer Science
Peking University
Beijing, China 100871
yuzf12@pku.edu.cn

**Tiejun Huang**
School of Computer Science
Peking University
Beijing, China 100871
tjhuang@pku.edu.cn

## Abstract

Spiking Neural Networks (SNNs) are promising in neuromorphic hardware owing to utilizing spatio-temporal information and sparse event-driven signal processing. However, it is challenging to train SNNs due to the non-differentiable nature of the binary firing function. The surrogate gradients alleviate the training problem and make SNNs obtain comparable performance as Artificial Neural Networks (ANNs) with the same structure. Unfortunately, batch normalization, contributing to the success of ANNs, does not play a prominent role in SNNs because of the additional temporal dimension. To this end, we propose an effective normalization method called temporal effective batch normalization (TEBN). By rescaling the presynaptic inputs with different weights at every time-step, temporal distributions become smoother and uniform. Theoretical analysis shows that TEBN can be viewed as a smoother of SNN's optimization landscape and could help stabilize the gradient norm. Experimental results on both static and neuromorphic datasets show that SNNs with TEBN outperform the state-of-the-art accuracy with fewer time-steps, and achieve better robustness to hyper-parameters than other normalizations.

## 1 Introduction

As the third generation of neural network models, Spiking Neural Networks (SNNs) are closer to the neuronal circuits of the brain and have recently received substantial attention in artificial intelligence and neuroscience [32, 40]. SNNs process spatio-temporal information with asynchronous discrete

---

[*]These authors contributed equally to this work.
[†]Corresponding author

36th Conference on Neural Information Processing Systems (NeurIPS 2022).

events [25, 52, 53, 54], which is ultra-low-power and highly compatible with neuromorphic and FPGA devices. However, due to the non-differentiable nature of the membrane potential at the firing time, it remains challenging to train high-performance deep SNNs [29, 44].

There are two main methodologies to address the training problems. One is the conversion-based method, which converts pre-trained Artificial Neural Networks (ANNs) to SNNs by converting ReLU activation to spiking neurons and retaining the weights [5, 36, 43]. The converted SNNs usually require a large number of time-steps to achieve comparable performance as source ANNs [4, 18], and can not be applied to neuromorphic datasets [9]. The second is end-to-end training with back-propagation, which is promising as it only needs a few time-steps [49]. In the end-to-end training, the forward path is achieved with a non-differentiable spiking function, and the network back-propagates through surrogate gradient functions [35] or temporal dependencies [35, 56].

Batch normalization (BN) contributes to the success of ANNs by flattening the loss landscape and reducing the internal covariate shift (ICS) [24, 42]. Whereas when it comes to SNNs, the existing normalization methods rarely take full advantage of the additional temporal dimension, which restricts the performance of SNNs [50]. Furthermore, the temporal dynamics of spiking neuron is tightly in connection with the neuronal parameters. Therefore, when these parameters are changed, the performance of some methods may be degraded [13].

In this paper, we propose Temporal Effective Batch Normalization (TEBN), a novel and effective normalization approach that regularizes the temporal distribution through weighting the presynaptic inputs of the spiking neurons. We prove that our approach could smooth the optimization landscape of SNN and help stabilize the gradient norm. The experiment results on CIFAE-10, CIFAR-100, and DVS-CIFAR10 datasets show that our approach achieves better classification performance and robustness to hyper-parameters than other BN methods. Besides, the proposed method outperforms the state-of-the-art learning methods, using fewer time-steps.

## 2 Related Works

### 2.1 Learning of Spiking Neural Networks

In recent years, many learning methods have been developed to train deep SNNs with excellent performance, which can be classified into ANN to SNN conversion (ANN2SNN) [4, 5, 9, 11, 17, 18, 20, 22, 33, 41, 43, 45] and end-to-end backpropagation methods [6, 7, 12, 16, 21, 29, 29, 31, 35, 44, 49, 57]. ANN2SNN pre-trains a source ANN and then converts it to an SNN by changing the artificial neuron model to the spiking neuron [5]. The main idea is to use the firing rates [18] or average postsynaptic potentials [9] of spiking neurons to approximate the ReLU activations of artificial neurons. Although some advanced conversion methods have achieved almost loss-less accuracy on large-scale datasets with ResNet and VGG-16 structures [4, 9, 18, 43], they overlook the rich temporal dynamic characteristics of SNNs and demand many time steps to approach the accuracy of pre-trained ANNs. In contrast, the backpropagation method directly trains SNNs by unfolding the network over the time-steps and then computing the gradient on both spatial and temporal domains, which borrows the idea of training Recurrent Neural Networks (RNNs) with backpropagation through time [49]. As the gradient of spikes with respect to membrane potential is non-differentiable, the surrogate gradient is proposed by approximating the gradient with smooth functions [35, 55]. The backpropagation method with surrogate gradients can be applied to both static and neuromorphic datasets, and requires far fewer time-steps than ANN2SNN methods. Another kind of backpropagation method is timing-based [2, 8, 26, 56], which directly computes the gradient of the timing of firing spikes to the membrane potential and does not require unfolding the network over the time-steps. However, they are usually limited to shallow networks that are less than 15 layers [3].

### 2.2 Normalization in Spiking Neural Networks

The training of well-behaved ANNs is made possible by using normalization approaches, including batch normalization [24], group normalization [51], and layer normalization [1]. Therefore, the researchers try to incorporate normalization approaches in ANNs to enhance the learning of SNNs. For example, NeuNorm [50] constructed auxiliary feature maps receiving the lateral inputs from the same layer to control the strength of the stimulus, which normalized the data along the channel dimension. Recently, some methods have modified batch normalization from a temporal perspective

to fit the training of SNNs. Zheng et al. [57] proposed a threshold-dependent batch normalization (tdBN) method, which extended the scope of BN to the additional temporal dimension. tdBN is stated to be able to alleviate the vanishing or explosion of gradients. Just as vanilla BN, the utilization of shared parameters may neglect the negative impact brought by the unusual temporal distributions. PSP-BN [23] used unique statistics, the second raw moment, as the denominator in normalization and it can be inserted right after the spiking functions. BNTT [27] regulated the firing rates by utilizing separate sets of BN parameters on different time-steps, and can be used for learning in diverse scenarios [47]. The complexity of BN parameters is higher compared to other BN methods and could break the temporal coherence of information. Inspired by these methods, we aim to develop a modified BN method with low complexity of parameters and take advantage of the temporal distribution of presynaptic inputs in the meantime.

## 3 Preliminary

### 3.1 Spiking Neuron Model

Unlike traditional ANNs, SNNs use binary spike trains to transmit information. Here we consider the widely used Leaky-Integrate-and-Fire (LIF) [15] model to emulate the dynamics of spiking neurons, which is formulated as a first-order differential equation.

$$\tau_m \frac{d\boldsymbol{u}_i(t)}{dt} = -\boldsymbol{u}_i(t) + R\boldsymbol{I}_i(t), \tag{1}$$

where $\boldsymbol{u}_i(t)$ denotes the membrane potential of the $i$-th neuron at time $t$, $\tau_m$ represents the membrane time constant, $R$ and $I_i(t)$ denote the linear resistor and the input current to the $i$-th neuron at time $t$. For numerical simulations of LIF neurons, we need to consider a discrete version of the parameters dynamics. Similar to [49], the membrane potential $\boldsymbol{u}_i[t]$ of the $i$-th neuron at time-step $t$ is represented as:

$$\boldsymbol{u}_i[t] = \tau\boldsymbol{u}_i[t-1] + \sum_{j \in \mathrm{pre}(i)} \boldsymbol{w}_{ij}\boldsymbol{o}_j[t]. \tag{2}$$

Here $j$ denotes the index of pre-synaptic neuron and $\boldsymbol{o}_j[t]$ denotes the binary spike of neuron $j$ at time-step $t$, which equals 1 if there is a spike and 0 otherwise. The membrane potential $\boldsymbol{u}_i[t]$ decreases with a leak factor $\tau$ that is no more than 1, and increases with the summation of inputs spikes from the pre-synaptic neurons $\mathrm{pre}(i)$ through synaptic weight $\boldsymbol{w}_{ij}$. When the membrane potential $\boldsymbol{u}_i[t]$ exceeds a specific threshold $\theta$, the neuron will fire a spike and then the membrane potential is reset to 0. By combining the sub-threshold dynamics (Eq. 2) and hard reset mechanism, the whole iterative LIF model can be determined by:

$$\boldsymbol{u}_i[t] = \tau\boldsymbol{u}_i[t-1](1 - \boldsymbol{o}_i[t-1]) + \sum_{j \in \mathrm{pre}(i)} \boldsymbol{w}_{ij}\boldsymbol{o}_j[t], \tag{3}$$

$$\boldsymbol{o}_i[t] = H(\boldsymbol{u}_i[t] - \theta), \tag{4}$$

where $H$ denotes the Heaviside step function.

### 3.2 Batch Normalization

Batch Normalization (BN) has been a typical strategy for training deep neural networks. By fixing the distribution of inputs, BN can reduce the Internal Covariate Shift (ICS) during training, allowing stable convergence of deeper neural networks [24]. Considering an analog neuron with input $\boldsymbol{x} = (\boldsymbol{x}^{(1)}, \boldsymbol{x}^{(2)}, \cdots, \boldsymbol{x}^{(m)})$, where $m$ is a mini-batch size. BN first normalizes each activation with:

$$\hat{\boldsymbol{x}}^{(b)} = \frac{\boldsymbol{x}^{(b)} - \boldsymbol{\mu}}{\sqrt{\boldsymbol{\sigma}^2}}, \quad b = 1, 2, ..., m, \tag{5}$$

where the expectation $\boldsymbol{\mu}$ and the variance $\boldsymbol{\sigma}^2$ are computed by $\boldsymbol{\mu} = \frac{1}{m}\sum_{b=1}^{m}\boldsymbol{x}^{(b)}$ and $\boldsymbol{\sigma} = \frac{1}{m}\sum_{b=1}^{m}(\boldsymbol{x}^{(b)} - \boldsymbol{\mu})^2$, respectively. To enhance the representation capability and ensure BN can represent the identity transformation, the learnable parameters $\gamma$ and $\beta$ are introduced to scale and shift the normalized vector, and the output of BN is:

$$BN(\boldsymbol{x}^{(b)}) = \gamma\hat{\boldsymbol{x}}^{(b)} + \beta. \tag{6}$$

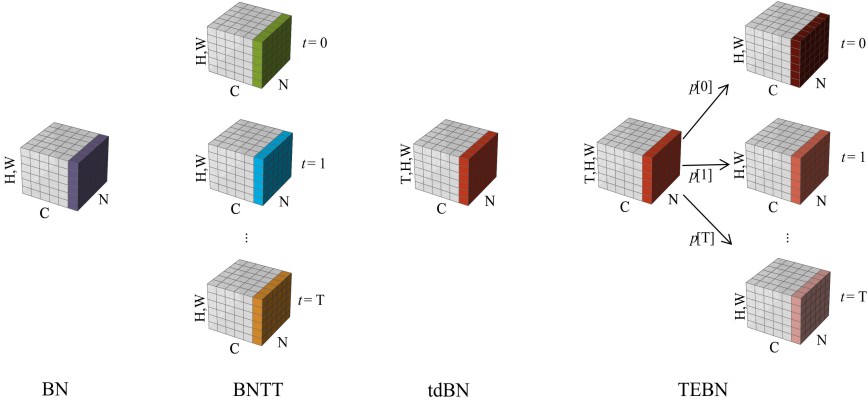

Figure 1: Illustrations of different BN methods for SNNs.

## 4 Methodology

### 4.1 Covariate Shift in the Temporal Dimension

ICS has been widely regarded as the key to the effectiveness of BN. In SNNs, synaptic currents are fed into spiking neurons sequentially, where BN is still the key to enforcing effective SNN training. Intuitively, the spikes triggering asynchronous currents may strengthen the internal covariate shift in the temporal domain. We refer to this phenomenon as Temporal Covariate Shift (TCS), which can be viewed as a special case of ICS. Default BN actually updates its statistics at each time-step, approximating the statistics of total time-step inputs. Here we build the relationship between the expectation and variance across all time-steps $\left(\boldsymbol{\mu}_{total}, \boldsymbol{\sigma}_{total}^2\right)$ and those of the single time-step $\left(\boldsymbol{\mu}[t], \boldsymbol{\sigma}^2[t]\right)$ $(t = 1, 2, \cdots, T)$:

$$\boldsymbol{\mu}_{total} = \frac{1}{T} \sum_{i=1}^{T} \boldsymbol{\mu}[i], \tag{7}$$

$$\boldsymbol{\sigma}_{total}^2 = \frac{1}{T} \sum_{i=1}^{T} \boldsymbol{\sigma}^2[i] + \frac{1}{T}(1 - \frac{1}{T}) \sum_{i=1}^{T} \boldsymbol{\mu}^2[i] - \frac{2}{T^2} \sum_{i \neq j} \boldsymbol{\mu}[i]\boldsymbol{\mu}[j]. \tag{8}$$

By supposing that $\boldsymbol{\mu}[t]$ is approximately identical for each time-step $t$, we get:

$$\hat{\boldsymbol{\sigma}}_{total}^2 = \frac{1}{T} \sum_{i=1}^{T} \boldsymbol{\sigma}^2[i]. \tag{9}$$

Hence, the Gaussian distribution across all time-steps can be seen as the composition of $T$ independent distributions. Fig. 1 illustrates three typical BN methods used in SNN, including default BN [24], BNTT [27], and tdBN [57]. The statistics and parameters these methods use are listed in Tab. 1.

Table 1: Statistics and parameters of presynatic inputs with different BN methods

| | t=1 | t=2 | ... | t=T |
|---|---|---|---|---|
| BNTT | $\boldsymbol{\mu}[1], \boldsymbol{\sigma}^2[1]$ | $\boldsymbol{\mu}[2], \boldsymbol{\sigma}^2[2]$ | | $\boldsymbol{\mu}[T], \boldsymbol{\sigma}^2[T]$ |
| | $\boldsymbol{\gamma}[1]$ | $\boldsymbol{\gamma}[2]$ | | $\boldsymbol{\gamma}[T]$ |
| tdBN | | $\boldsymbol{\mu}_{total}, \boldsymbol{\sigma}_{total}^2$ | | |
| | | $\boldsymbol{\gamma}, \boldsymbol{\beta}$ | | |
| TEBN | | $\boldsymbol{\mu}_{total}, \boldsymbol{\sigma}_{total}^2$ | | |
| | $\hat{\boldsymbol{\gamma}}[1], \hat{\boldsymbol{\beta}}[1]$ | $\hat{\boldsymbol{\gamma}}[2], \hat{\boldsymbol{\beta}}[2]$ | | $\hat{\boldsymbol{\gamma}}[T], \hat{\boldsymbol{\beta}}[T]$ |

As shown in Tab. 1, BNTT exploits separate sets of mean, variance, and the scale and shift parameters to model the distribution of presynaptic input at each time-step. It establishes a delicate model of distributions while ignoring the time dependence of the input spikes. In addition, BNTT takes

large amounts of computation in implementation as it needs the same number of BN modules as the time-step. Different from it, tdBN calculates the overall expectation and variance for all T time-steps, and the scale and shift parameters are also shared for all time-steps. The difference between tdBN and default BN lies in the implementation. As illustrated in Fig. 1, the input statistics of tdBN consider the statistics of all the batches and time-steps, while default BN uses the momentum update to approximate the total statistics. The merit of using $\boldsymbol{\mu}_{total}, \boldsymbol{\sigma}^2_{total}$ in tdBN is that when doing backpropagation, there is only one set of input statistics that may contribute to the alignment of temporal distributions. Actually, tdBN considers the time dependence of the input spikes, but ignores the varying temporal characteristics at different time-steps.

## 4.2 Temporal Effective Batch Normalization

To model the temporal distributions in SNN training without using too many parameters, combining the findings of Eq. 9, we treat the distribution of presynaptic currents at each time step as a scaling of the overall distribution. To this end, we propose Temporal Effective Batch Normalization (TEBN), which assigns different weights $p$ to each time-step to distinguish their effect on the final result. The weights are learnable to find an optimum for every time-step to implement efficient inference. Given that $\boldsymbol{x}^{l-1}[t]$ is the input to layer $l$ at time-step $t$, the spiking neuron with TEBN can be described as:

$$\boldsymbol{x}^{l-1}[t] = \boldsymbol{W}^l \boldsymbol{o}^{l-1}[t], \tag{10}$$

$$\boldsymbol{u}^l[t] = \tau \boldsymbol{u}^l[t-1](1 - \boldsymbol{o}^l[t-1]) + \hat{\boldsymbol{x}}^{l-1}[t], \tag{11}$$

$$\text{where} \quad \hat{\boldsymbol{x}}[t] = \text{TEBN}(\boldsymbol{x}[t]) = \hat{\boldsymbol{\gamma}}[t] \frac{\boldsymbol{x}[t] - \boldsymbol{\mu}}{\sqrt{\boldsymbol{\sigma}^2 + \epsilon}} + \hat{\boldsymbol{\beta}}[t], \tag{12}$$

$$\hat{\boldsymbol{\gamma}}[t] = \boldsymbol{\gamma} \times p[t], \ \hat{\boldsymbol{\beta}}[t] = \boldsymbol{\beta} \times p[t]. \tag{13}$$

Here $\boldsymbol{u}^l[t]$ and $\boldsymbol{o}^l[t]$ denote the membrane potential and binary output spikes of all neurons in layer $l$ at time-step $t$. $\boldsymbol{W}^l$ is the synaptic weights between layer $l-1$ and layer $l$, and $\hat{\boldsymbol{x}}^l[t]$ represents the output of TEBN in layer $l$ at time-step $t$. We calculate the mean $\boldsymbol{\mu}$ and variance $\boldsymbol{\sigma}^2$ from samples of all time-steps. $\epsilon$ is a small constant, which is added to guarantee numerical stability. Each TEBN layer processes time-invariant BN parameters $\boldsymbol{\gamma}$ and $\boldsymbol{\beta}$ and learnable weight parameters $p[t]$. Since the parameters $p[t](t = 0, 1, ..., T)$ are self-adapting to adjust the distribution of $\boldsymbol{x}[t]$, TEBN will help relieve the problem of TCS.

## 4.3 Detailed Learning Rule in SNNs

We apply TEBN to SNNs and derive the detailed learning rule. Actually, the problem is to compute the gradients of the loss function $\mathcal{L}$ with respect to the weights $\partial \boldsymbol{W}^l_{ij}$ and with respect to the parameter $p[t]$, where $\partial \boldsymbol{W}^l_{ij}$ denotes the weight between the $j$-th neuron in layer $l-1$ and the $i$-th neuron in layer $l$. Similar to the previous works [27, 57], we compute the gradient by unfolding the network over the simulation time-steps, and get:

$$\frac{\partial \mathcal{L}}{\partial \boldsymbol{u}^l_i[t]} = \frac{\partial \mathcal{L}}{\partial \boldsymbol{o}^l_i[t]} \frac{\partial \boldsymbol{o}^l_i[t]}{\partial \boldsymbol{u}^l_i[t]} + \frac{\partial \mathcal{L}}{\partial \boldsymbol{u}^l_i[t+1]} \frac{\partial \boldsymbol{u}^l_i[t+1]}{\partial \boldsymbol{u}^l_i[t]} \tag{14}$$

$$\frac{\partial \mathcal{L}}{\partial W^l_{ij}} = \frac{\partial \mathcal{L}}{\partial \boldsymbol{x}^{l-1}_i[t]} \boldsymbol{o}^{l-1}_j[t] = \frac{\partial \mathcal{L}}{\partial \boldsymbol{u}^l_i[t]} \frac{\partial \hat{\boldsymbol{x}}^{l-1}_i[t]}{\partial \boldsymbol{x}^{l-1}_i[t]} \boldsymbol{o}^{l-1}_j[t] \tag{15}$$

where $\boldsymbol{u}^l_i[t]$ and $\boldsymbol{o}^l_i[t]$ denote the membrane potential and output spikes of the $i$-th neurons in layer $l$ at time-step $t$, respectively. $\boldsymbol{x}^l_i[t]$ and $\hat{\boldsymbol{x}}^l_i[t]$ represent the $i$-th input and output of TEBN in layer $l$ at time-step $t$. As the derivative of spike with respect to the membrane potential $\frac{\partial \boldsymbol{o}^l_i[t]}{\partial \boldsymbol{u}^l_i[t]}$ is non-differentiable, the surrogate gradient method [35, 49] is used to smooth the Heaviside step function.

The weight parameter $p[t]$ allows the network to capture temporal dynamics when training synaptic connections, and the gradient of the loss function $\mathcal{L}$ to the weight parameter $p^l[t]$ in layer $l$ is:

$$\frac{\partial \mathcal{L}}{\partial p^l[t]} = \sum_i \frac{\partial \mathcal{L}}{\partial \boldsymbol{u}^l_i[t]} \frac{\partial \boldsymbol{u}^l_i[t]}{\partial p^l[t]} = \sum_i \left[ \frac{\partial \mathcal{L}}{\partial \boldsymbol{u}^l_i[t]} \left( \gamma^l_i \frac{\boldsymbol{x}^{l-1}_i[t] - \boldsymbol{\mu}^l_i}{\sqrt{(\boldsymbol{\sigma}^l_i)^2 + \epsilon}} + \beta^l_i \right) \right], \tag{16}$$

where $\boldsymbol{\mu}^l_i$ and $(\boldsymbol{\sigma}^l_i)^2$ denote the mean and variance of the $i$-th neuron from samples of all time-steps. $\gamma^l_i$ and $\beta^l_i$ are the $i$-th parameters of TEBN.

## 4.4 Related to Learnable Input Resistance of LIF Neuron

Like Long Short-Term Memory (LSTM) [19], the LIF neuron has a similar function in controlling the memory. As illustrated in Eq. 1, the LIF neuron is able to remember current input information and forget some information from the past, which is under the regulation of the relative scale of membrane time constant $\tau_m$ and linear resistor $R$. A larger $\tau_m$ or a smaller $R$ typically indicates a more reinforced memory. The resistor is usually regarded as a latent variable absorbed in the weights (c.f. Eq. 2). Fixing the time constant has its problem. First, when given the time-step number for inference, the LIF neuron cannot maintain a memory to enhance the internal representation. Second, the accuracy of SNN is subjected to the chosen $\tau$ [13]. In this regard, our TEBN uses learnable hyperparameter $p[t]$ to rescale the presynaptic inputs at time-step $t$, which can be regarded as optimizing a learnable input resistance $R$. The learnable $p[t]$ can share the benefit of the learnable $\tau$ proposed in [13] to improve the representation. Different from [13], the time-varying characteristic further enables the network to find an optimal proportion of remembrance to input at each time-step, which endows the network with better fitting ability and robustness to hyperparameters.

## 5 Smoothing Optimization: A Theoretical Analysis

The proposed normalization methods suggest diverse disposal for different time-steps of neurons. The mini-batch training makes each neuron $i$ have a dimension of mini-batches. We will show in our analysis that the special design of TEBN can not only flatten the loss landscape with regard to the input signal, but also help absorb the decay of gradient norm of mini-batches in adjacent time-steps.

**Theorem 1.** *For the time-dependent input signal $\boldsymbol{x}_i[t]$ with $t$ as the current time-step and $i$ as the neuron index, the relation between the gradient of $\nabla \mathcal{L}$ w.r.t. $\boldsymbol{x}_i[t]$ in a TEBN network $\nabla_{\boldsymbol{x}_i[t]}\mathcal{L}(\hat{\boldsymbol{x}}[t])$ and the corresponding gradient $\nabla_{\boldsymbol{x}_i[t]}\tilde{\mathcal{L}}(\boldsymbol{x}[t])$ in a non-TEBN counterpart network which has a loss $\tilde{\mathcal{L}}$ is as follows:*

$$\left\| \nabla_{\boldsymbol{x}_i[t]}\mathcal{L} \right\| \leqslant \frac{\hat{\boldsymbol{\gamma}}_i[t]}{\boldsymbol{\sigma}_i} \left\| \nabla_{\boldsymbol{x}_i[t]}\tilde{\mathcal{L}} \right\| \tag{17}$$

*where $\hat{\boldsymbol{x}}[t] = TEBN(\boldsymbol{x}[t])$, $\hat{\boldsymbol{\gamma}}_i[t]$ is the $i$-th element of $\hat{\boldsymbol{\gamma}}[t]$ in TEBN. Note that the $L_2$ norm is calculated in the dimension of mini-batches.*

The Lipschitzness of the loss with TEBN blocks is shown to exploit a better Lipschitz constant in Theorem 1, which contributes to the restraint of the amplification of gradient update steps. Theorem 1 implies that the TEBN block performs as traditional BN in ANNs. TEBN actually smooths the gradient of the temporal-varying currents. With a favored optimization landscape, SNNs can also have the opportunity to flatten the optimization. Moreover, as the existence of the time-dependent $\hat{\boldsymbol{\gamma}}_i[t]$, the bound of the Lipschitz constant is non-static. By setting $p[t]$s equal 1, TEBN degenerates to traditional BN. This may decrease the capability of controlling the temporal Lipschitz constant in practice.

As illustrated in Sec. 4.4, the TEBN block scaling presynaptic inputs can be seen as learnable resistance in SNN. We further check the Lipschitzness of the gradient between time-steps $t-1$ and $t$. Our analysis can help understand why extra scaling on the time-step is necessary.

**Theorem 2.** *For the time-dependent input signal $\boldsymbol{x}_i[t]$ with $t$ as the current time-step and $i$ as the neuron index, denote $\bar{\boldsymbol{x}}_i^{l-1}[t]$ as the hidden output of TEBN without scaling presynaptic inputs by $p^l[t]$. Then, the relation between the gradient of $\nabla\mathcal{L}$ w.r.t. $\bar{\boldsymbol{x}}_i^{l-1}[t]$ and $\bar{\boldsymbol{x}}_i^{l-1}[t-1]$ is as follows:*

$$\|\nabla_{\bar{\boldsymbol{x}}_i^{l-1}[t-1]}\mathcal{L}\| \leq \frac{p^l[t-1]}{p^l[t]}\tau\sqrt{k}\left(1+\theta\hbar_{\max}\right)\|\nabla_{\bar{\boldsymbol{x}}_i^{l-1}[t]}\mathcal{L}\| + p^l[t-1]\hbar_{\max}\|\nabla_{\boldsymbol{o}_i^l[t-1]}\mathcal{L}\|, \tag{18}$$

*where $\hbar_{\max}$ is the maximum that the surrogate gradient function can take, and $k$ is the mini-batch size.*

Theorem 2 depicts the magnification of the gradient norm with regard to time. When setting all $p[t]$s equal one, TEBN can provide the effects of traditional BN. The Lipschitz constant of gradients for adjacent time is constant $\tau\sqrt{k}\left(1+\theta\hbar_{\max}\right)$, which may cause the norm to be unstable while training. The use of trainable $p[t]$, in this regard, can help alleviate this situation and stabilize the gradient norm, which is essential for a training procedure.

To summarize, directly applying the traditional resolution of Batch Norm into SNN can benefit the training process: smoothing the optimization landscape. We conclude from Theorem 1 that our proposed normalization method has at least the same ability to preserve gradient norm. Besides, we also discover that the Lipschitz constant of BN depends on some neuron parameters like the leak factor $\tau$ and the threshold $\theta$ in Theorem 2. We believe a trainable scaling factor in the temporal dimension will mitigate the gradient norm from explosion and vanishing.

## 6 Experiments

In this section, we validate the effectiveness of our proposed TEBN for classification tasks on both static and neuromorphic datasets. We first compare our algorithm with BN methods to demonstrate the advantages of TEBN. Further, we compare our method with other state-of-the-art approaches. Finally, we visualize the distribution of TEBN outputs and demonstrate that the proposed method is robust to hyper-parameters. We follow the same data pre-processing protocol as previous work [50]. More details of the configurations are provided in the supplementary.

### 6.1 Comparison with other BN Methods

We first evaluate the performance of the proposed TEBN and other normalization methods. For a fair comparison, we do not use advanced data augmentation like cutout [10]. Tab. 2 reports the test accuracy on both traditional static CIFAR-10, CIFAR-100 [28] datasets and neuromorphic DVS-CIFAR10 [30] dataset. On the CIFAR-10 dataset, our TEBN achieves 94.57% top-1 accuracy with ResNet-19 network using only 2 time-steps. When the network structure is the same, TEBN outperforms the other BN methods, even with fewer time-steps T. We get the same conclusion on other datasets. For example, on CIFAR-100, our method achieves better performance (74.37% v.s. 66.6%) and fewer time-steps (4 v.s. 50) than BNTT [27]. On DVS-CIFAR10, the accuracy of our method is 14.6% (75.1% v.s. 60.5%) and 11.9% (75.1% v.s. 63.2%) higher than NeuNorm and BNTT, which use 4 times and 2 times as many simulation time-steps T, respectively. Moreover, we achieve better performance (64.29% v.s. 63.72%) and fewer time-steps (4 v.s. 6) than tdBN [57] on ImageNet. All the accuracies of other methods reported in the tables result from the literature.

Table 2: Comparisons with different normalization methods.

| Dataset | Model | Methods | Architecture | Time-steps | Accuracy(%) |
|---------|-------|---------|--------------|------------|-------------|
| CIFAR-10 | SPIKE-NORM [43] | ANN2SNN | VGG-16 | 2500 | 91.55 |
| | NeuNorm [50] | Surrogate Gradient | CIFARNet | 12 | 90.53 |
| | BNTT [27] | Surrogate Gradient | VGG-9 | 20 | 90.30 |
| | tdBN [57] | Surrogate Gradient | ResNet-19 | 6 | 93.16 |
| | | | | 4 | 92.92 |
| | | | | 2 | 92.34 |
| | **TEBN** | Surrogate Gradient | VGG-9 | 4 | 92.81 |
| | | | ResNet-19 | 6 | **94.71** |
| | | | | 4 | **94.70** |
| | | | | 2 | **94.57** |
| CIFAR100 | SPIKE-NORM [43] | ANN2SNN | VGG-16 | 2500 | 70.90 |
| | BNTT [27] | Surrogate Gradient | VGG-11 | 50 | 66.60 |
| | **TEBN** | Surrogate Gradient | VGG-11 | 4 | 74.37 |
| | | | ResNet-19 | 6 | **76.41** |
| | | | | 4 | **76.13** |
| | | | | 2 | **75.86** |
| DVS-CIFAR10 | NeuNorm [50] | Surrogate Gradient | 7-layer CNN | 40 | 60.50 |
| | BNTT [27] | Surrogate Gradient | 7-layer CNN | 20 | 63.20 |
| | tdBN [57] | Surrogate Gradient | ResNet-19 | 10 | 67.80 |
| | **TEBN** | Surrogate Gradient | 7-layer CNN | 10 | **75.10** |
| ImageNet | SPIKE-NORM [43] | ANN2SNN | ResNet-34 | 2500 | 69.96 |
| | tdBN [57] | Surrogate Gradient | ResNet-34 | 6 | 63.72 |
| | **TEBN** | Surrogate Gradient | ResNet-34 | 4 | 64.29 |

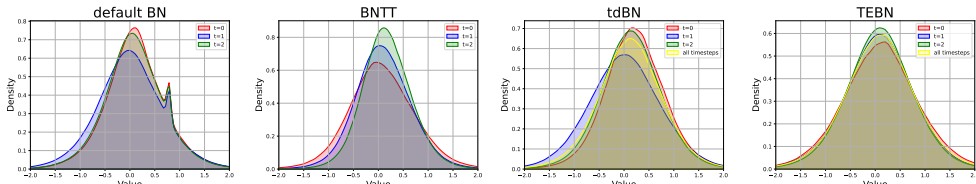

Figure 2: Distribution of presynaptic inputs in layer 2 with different BN methods on CIFAR-10.

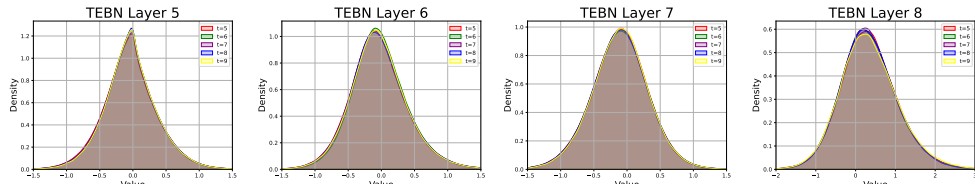

Figure 3: Distribution of presynaptic inputs with TEBN on DVS-CIFAR10

## 6.2 Comparison with the State-of-the-Art

We compare our method with other state-of-the-art learning methods for SNNs on CIFAR-10, CIFAR-100, and DVS-CIFAR10 datasets and report the results in Tab. 3. With our reproduction of the CNN model, our TEBN outperforms HC ( 93.96% v.s. 92.94%, VGG-11) and TSSL-BP (92.65% v.s. 91.41%, 7-layer CNN) with fewer time-steps. We contribute this to the better representational capacity of TEBN. On CIFAR-100, our TEBN achieves an accuracy of 75.86% with 2 time-steps on ResNet-19, which is better than results of ANN2SNN and hybrid training. On DVS-CIFAR10, our method performs better than PLIF and TET using the same architecture. Besides, we also adopt the cutout augmentation on static datasets denoted by "*" in the table. Details of the data augmentation are presented in the supplementary. Cutout helps TEBN to realize better performance. Compared with other surrogate gradient methods such as TET and Dspike, our TEBN model performs consistently better. For instance, We achieve the top-1 accuracy of 78.76% on CIFAR-100 dataset with ResNet-19 (T=6), which is 4.04% higher than that of TET with the same network structure and time-steps. Moreover, our method outperforms the state-of-the-art learning methods on ImageNet dataset. All the accuracies of other methods reported in the tables result from the literature.

## 6.3 Distribution of TEBN Outputs

We train 9-layer CNN models to visualize the temporal distribution of normalization layers with default BN, BNTT, tdBN, and the proposed TEBN. Fig. 2 illustrates the results of inputs in layer 2 on CIFAR-10 with 3 time-steps. The default BN shows an irregular shape instead of a standard normal distribution. On the other hand, BNTT, tdBN, and TEBN reshape the presynaptic inputs into a bell-shaped distribution. Among the three methods, the temporal distribution of TEBN appears more homogeneous. Unlike the static datasets, neuromorphic datasets contain rich temporal information, which means presynaptic inputs in different time-steps may have varied distributions. Here we train 9-layer CNN model on DVS-CIFAR10 with TEBN and visualize the temporal distribution of the normalization layers in Fig. 3. The last five time-steps are selected for better visual effects. All the chosen time-steps have an almost identical distribution. We can conclude that our TEBN can play a role in adjusting the current distribution of DVS data.

## 6.4 Robustness on Hyper-parameters

Previous work has revealed that improper initial value of membrane time constant $\tau$ will hamper the learning of SNN [13]. Here we compare the performances of different BN methods with various $\tau$. We train 9-layer CNN models on CIFAR-10 dataset with 2 time-steps and $\tau$ values chosen from [0.1, 0.25, 0.5, 0.75, 1.0]. As shown in Fig. 4(a), the models with TEBN maintain stable accuracy when $\tau$ changes, whereas others fluctuate. The result indicates the robustness of our method to various $\tau$.

In fact, the parameters $p[t]$ are self-adapted to find the optimum when $\tau$ changes. We conduct additional experiments with TEBN. The time-step is set to 6, and $\tau$ varies from 0.25 to 1.0 with step

Table 3: Compare with state-of-the-art supervised training methods. * denotes using cutout.

| Dataset | Model | Methods | Architecture | Time-steps | Accuracy(%) |
|---|---|---|---|---|---|
| CIFAR-10 | RMP [18] | ANN2SNN | ResNet-20 | 2048 | 91.36 |
| | Opt. [9] | ANN2SNN | ResNet-20 | 128 | 93.56 |
| | PTL [48] | ANN2SNN | VGG-11 | 16 | 91.24 |
| | QCFSA [4] | ANN2SNN | ResNet-20 | 64 | 92.35 |
| | HC [37] | Hybrid Training | VGG-11 | 2500 | 92.94 |
| | TC [58] | Time-based Gradient | VGG-16 | - | 92.68 |
| | TSSL-BP [56] | Time-based Gradient | 7-layer CNN | 5 | 91.41 |
| | Dspike* [31] | Surrogate Gradient | ResNet-18 | 6 | 94.25 |
| | | | | 4 | 93.66 |
| | | | | 2 | 93.13 |
| | TET* [10] | Surrogate Gradient | ResNet-19 | 6 | 94.50 |
| | | | | 4 | 94.44 |
| | | | | 2 | 94.16 |
| | **TEBN** | Surrogate Gradient | 7-layer CNN | 4 | 92.65 |
| | | | VGG-11 | 4 | 93.96 |
| | | | ResNet-19 | 6 | **94.71** |
| | | | | 4 | **94.70** |
| | | | | 2 | **94.57** |
| | **TEBN*** | Surrogate Gradient | ResNet-19 | 6 | **95.60** |
| | | | | 4 | **95.58** |
| | | | | 2 | **95.45** |
| CIFAR100 | RMP [18] | ANN2SNN | ResNet-20 | 2048 | 67.82 |
| | Opt. [9] | ANN2SNN | ResNet-20 | 512 | 72.34 |
| | QCFSA [4] | ANN2SNN | ResNet-20 | 128 | 70.55 |
| | HC [37] | Hybrid Training | VGG-11 | 2500 | 70.94 |
| | Dspike* [31] | Surrogate Gradient | ResNet-18 | 6 | 74.24 |
| | | | | 4 | 73.35 |
| | | | | 2 | 71.68 |
| | TET* [10] | Surrogate Gradient | ResNet-19 | 6 | 74.72 |
| | | | | 4 | 74.47 |
| | | | | 2 | 72.87 |
| | **TEBN** | Surrogate Gradient | VGG-11 | 4 | 74.37 |
| | | | ResNet-19 | 6 | **76.41** |
| | | | | 4 | **76.13** |
| | | | | 2 | **75.86** |
| | **TEBN*** | Surrogate Gradient | ResNet-19 | 6 | **78.76** |
| | | | | 4 | **78.71** |
| | | | | 2 | **78.07** |
| DVS-CIFAR10 | PLIF [13] | Surrogate Gradient | 6-layer CNN | 20 | 74.80 |
| | Dsipke [31] | Surrogate Gradient | ResNet-18 | 10 | 75.40 |
| | TET [10] | Surrogate Gradient | VGGSNN | 10 | 83.17 |
| | **TEBN** | Surrogate Gradient | 6-layer CNN | 10 | 80.00 |
| | | | VGGSNN | 10 | **84.90** |
| ImageNet | SEW [12] | Surrogate Gradient | SEW ResNet-34 | 4 | 67.04 |
| | TET [10] | Surrogate Gradient | SEW ResNet-34 | 4 | 68.00 |
| | **TEBN** | Surrogate Gradient | SEW ResNet-34 | 4 | **68.28** |

0.25. As shown in Fig. 4(b)-4(c), smaller $\tau$ makes $p$ converge to a bigger value and vice versa. Note that when $\tau = 1.0$, the LIF neuron degenerates to the IF neuron, which affects the optimization of $p$.

# 7   Conclusion

An effective and robust normalization method TEBN is proposed for SNN in this paper. The main idea is utilizing time-specific learnable weights to rescale the presynaptic inputs, which serves to normalize the temporal information to smooth and conform temporal distribution. Optimizing the learnable parameters $p[t]$ can be regarded as adjusting the input resistance, which helps to enhance the fitting ability and robustness to specific hyper-parameters of spiking neurons. TEBN promotes the potential advantages of backpropagation methods for future practical applications of SNNs. In future researches, we aim to dive deeper into the temporal distribution of SNNs with BN and investigate

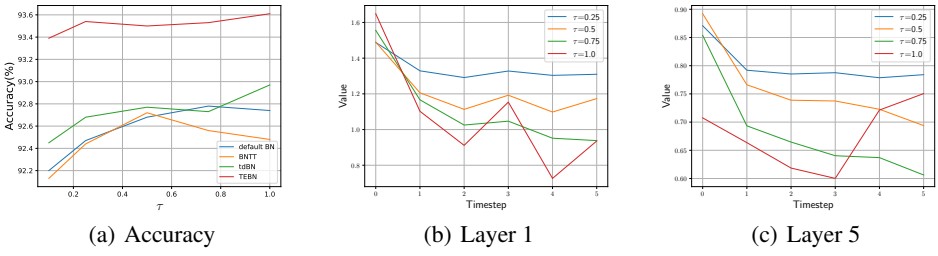

| (a) Accuracy | (b) Layer 1 | (c) Layer 5 |

Figure 4: The accuracy of different BN methods and $p[t]$ change with respect to $\tau$

more robust SNNs. Besides, recent works have shown that short-term plasticity (STP) [14, 39, 46] can be incorporated into ANNs to enhance efficiency and computational power [34, 38]. As STP performs a function of temporal filtering similar to TEBN, how to use the biologically-realistic filter-STP in SNNs is the future direction.

## 8 Acknowledgements

This work was supported by the National Natural Science Foundation of China Grants 62176003 and 62088102.

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
