# OpenReview forum: "Temporal Effective Batch Normalization in Spiking Neural Networks"
_NeurIPS.cc/2022/Conference — NeurIPS 2022 Accept_

### Official Review · Reviewer_Ljtf · 2022-06-20

**Rating:** 8
**Confidence:** 5
**Soundness:** 4 excellent
**Presentation:** 3 good
**Contribution:** 3 good

**Summary:**

This paper presents an efficient batch normalization (BN) method to smooth and uniform the temporal distributions of the presynaptic input in the Spiking Neural Networks (SNN). By combing the BN with temporal features, the proposed TEBN could alleviate the gradient vanishing problem to some extent. The proposed TEBN shows better classification accuracy and robustness to hyper-parameters on CIFAR-10, CIFAR-100 and DVS-CIFAR10 with fewer time-steps.

**Questions:**


1) Line 40, the sentence, ‘Theoretical analysis shows that our approach can be viewed as a smoother of SNN’s optimization landscape and could help stabilize the gradient norm.’ need a revision as it is repetitive with the statement in the abstract.
2) Line 65, please add a citation to support the statement (less than 15 layers).
3) Page9, Figures 2 and 3. The X and Y limit should be the same for both figures since the authors could compare distributions. Besides, I expect the author to conduct some quantitative measures to compare the distributions, rather than directly state that TEBN appears more homogeneous.
4) I would recommend the author validate the efficiency of mitigating gradient explosion and vanishing (Line 215) with the proposed TEBN using a deeper model (e.g., ResNet-50).


**Limitations:**

The efficiency of mitigating gradient explosion and vanishing (Line 215) with the proposed TEBN is not validated efficiently in this work. And what is the biological meaning of the hyper-parameters in LIF model? Or if the authors explain how different hyper-parameters influence TEBN with theoretical analysis not only experimental evaluations, the results would be more sound.

**Strengths And Weaknesses:**

Strengths:
1) The writing is clear and the motivation is clarified clearly. Besides, the theoretical grounding and experimental evaluation are also sufficient to show its originality and significance.
2) Experimental results also demonstrate the superiority of the proposed TEBN method. Through Fig. 1, we can see the advantages of the proposed TEBN Intuitively compared to tdBN.
3) This paper gives a detailed illustration of how TEBN processes in a surrogate-gradient SNN training method. If the authors show the input and output simultaneously with or without TEBN, the effectiveness maybe better in a visual way.
4) The spiking nondifferentiable leads to a non-smooth with the loss landscape, the authors analyze the smoothing optimization in sec. 5 which I think is very important for using the surrogate-gradient or ANN2SNN method training a deep SNN model.

Weaknesses:
How does internal covariate shift appear in SNN? In my opinion, because of the different scales from each layer's output, ICS in ANN  is obvious, however, in this paper, when you used the time-series LIF neuron model, the ICS should be changed in the time window. When using rate-based coding, the spikes are more intensively compared to temporal coding, how does TCS behave in addition to sec. 4.1?

---

> ### Author Response · Authors · 2022-08-02
> **Response to Reviewer Ljtf Part I/II**
>
> Thank you for your positive and constructive feedback.  We would like to address your concerns and answer your questions in the following.
>
> ***Comment 1.How does internal covariate shift appear in SNN? In my opinion, because of the different scales from each layer's output, ICS in ANN is obvious, however, in this paper, when you used the time-series LIF neuron model, the ICS should be changed in the time window. When using rate-based coding, the spikes are more intensively compared to temporal coding, how does TCS behave in addition to sec. 4.1?***
>
> We would like to point out that ICS in SNN behaves like ANN, that is, the overall distribution of current in different layers is different. This overall distribution is a combination of currents from different time steps. Since SNN adds a time dimension compared to ANN, the relationship between distributions in SNN is more complicated. Even in the same layer, the current distribution at different time steps is different, or in other words, shifted. Considering a layer of neurons, the spikes from the previous layer will cause different synaptic connections (i.e. weights) to be driven so that the current distributions will not be exactly the same. Therefore, TCS should appear less pronounced when using a more densely and uniform coding scheme, such as rate encoding.
>
> ***Comment 2. Line 40, the sentence, ‘Theoretical analysis shows that our approach can be viewed as a smoother of SNN’s optimization landscape and could help stabilize the gradient norm.’ need a revision as it is repetitive with the statement in the abstract.***
>
> Thanks for pointing it out. We have revised it to 'We prove that our approach could smooth the optimization landscape of SNN and help stabilize the gradient norm.' in the revised paper.
>
> ***Comment 3. Line 65, please add a citation to support the statement (less than 15 layers).***
>
> Thanks for your suggestion! We have added the reference [1] in the revised paper.
>
>
> ***Comment 4. Page9, Figures 2 and 3. The X and Y limit should be the same for both figures since the authors could compare distributions. Besides, I expect the author to conduct some quantitative measures to compare the distributions, rather than directly state that TEBN appears more homogeneous.***
>
> The reviewer noticed certain grammar and clarity issues. We have updated the manuscript accordingly. Furthermore, we have provided distribution figures of different methods with aligned axes in Sec. C of Supplement.
>
> In order to compare the distributions quantitatively, we plot the distributions shown in Fig. 2 as histograms, ranging from -1.5 to 1.5 with the bin of 0.05. Then we calculate the Kullback-Leibler(KL) divergence of the distribution histograms at every two timesteps. The results show that TEBN have smaller KL divergence, which means more homogeneous distributions.
>
> **Table R1: Comparison of KL divergence of distributions.**
> |Model | T=0，1 （×1e-3） |  T=0，2 （×1e-3） | T=1，2 （×1e-3）|
> |:-----:|:------:|:------:|:----:|
> | default BN |21.8|   0.9    |  16.9  |
> | BNTT[2] |17.1|   78.9    |  23.4  |
> | tdBN[3] |61.7|   6.6    |  40.3  |
> | TEBN |2.0|  6.0 |  1.1  |

---

> > ### Author Response · Authors · 2022-08-02
> > **Response to Reviewer Ljtf Part II/II**
> >
> > ***Comment 5. I would recommend the author validate the efficiency of mitigating gradient explosion and vanishing (Line 215) with the proposed TEBN using a deeper model (e.g., ResNet-50).***
> >
> > Due to limited time for rebuttal, here we add the experiment  of ResNet-34 on the ImageNet dataset to demonstrate the efficiency of our method. We first compare the performance of the proposed TEBN and tdBN [3]. As reported in Table R2, our method achieves better performance (64.29\% v.s. 63.72\%) and fewer time-steps  (4 v.s. 6) than tdBN [3]. Then we compare our method with other state-of-the-art learning methods [4,5] for SNNs. One can find that our method outperforms SEW [4] and TET [5] when the architecture and time-steps are the same. All these results demonstrate that the proposed method can apply to the complex dataset. We have added these results in Tables 2 and 3 of the revised paper.
> >
> > **Table R2: Comparison with other normalization method and the SOTA training methods on ImageNet dataset.**
> > |Model |Methods | Architecture | Time-steps| Accuracy(%) |
> > |:-----:|:------:|:------:|:----:|:-------:|
> > | tdBN[3] |Surrogate Gradient|   ResNet-34    |  6  |  63.72 |
> > | **TEBN** |Surrogate Gradient|   ResNet-34    |  4  |  **64.29** |
> > | SEW[4] |Surrogate Gradient|   SEW ResNet-34    |  4  |  67.04 |
> > | TET[5] |Surrogate Gradient|   SEW ResNet-34    |  4  |  68.00 |
> > | **TEBN** |Surrogate Gradient|  SEW ResNet-34 |  4  | **68.28**  |
> >
> >
> >
> > ***Comment 6.what is the biological meaning of the hyper-parameters in LIF model? Or if the authors explain how different hyper-parameters influence TEBN with theoretical analysis not only experimental evaluations, the results would be more sound.***
> >
> > In terms of mimicking the brain network, there are spectrums of models varying in their biological realistic and computational efficiency. The Integrate-and-Fire model (IF) is the simplest in biology and the most efficient for computation. There exist more complex models like the Leaky integrate-and-fire model (LIF), spiking response model(SRM), Hodgkin–Huxley model (HH), etc [6].
> > There are two hyper-parameters in LIF neuron we used, membrane time constant $\tau$ and firing threshold $\theta$. LIF neurons are able to remember current input information and forget some information from the past, which is under the regulation of the relative scale of membrane time constant $\tau$ . And a suitable threshold $\theta$ could maintain suitable firing rates and reduce information loss [3].
> >
> > Experimentally, we have presented the analysis of different time constants and firing thresholds to test the generalization. We showed the impacts of changing $\tau$ in Sec 6.4, where $\tau$=0.1, 0.25, 0.5, 0.75, 1.0. The accuracies of these $\tau$ settings are 93.39, 93.54, 93.5, 93.53, and 93.61. Besides, we test the effect of different thresholds. The firing threshold is a hyper-parameter of the spiking neuron, corresponding to a biological neuron characteristic. When the Threshold=[0.5, 1.0, 1.5], we obtain the Accuray=[92.22%, 92.57%, 92.99%] in experiments, as shown in Table R3. Overall, our experimental results show that our TEBN generalizes well when changing the hyper-parameters.
> >
> > **Table R3: Comparison with different thresholds on CIFAR-10 dataset.**
> > |Threshold |Accuracy(%)|
> > |:-----:|:------------:|
> > | 0.5 |   92.22 |
> > | 1.0 |   92.57 |
> > | 1.5 |   92.99 |
> >
> >
> >
> > [1] Tong Bu, Jianhao Ding, Zhaofei Yu, and Tiejun Huang. Optimized potential initialization for low-latency spiking neural networks. In In Proceedings of the AAAI Conference on Artificial Intelligence(AAAI), 2022
> >
> > [2]Youngeun Kim and Priyadarshini Panda. Revisiting batch normalization for training low-latency deep spiking neural networks from scratch. Frontiers in Neuroscience, 15:773954–773954, 2021.
> >
> > [3]Hanle Zheng, Yujie Wu, Lei Deng, Yifan Hu, and Guoqi Li. Going deeper with directly trained larger spiking neural networks. In Proceedings of the AAAI Conference on Artificial Intelligence(AAAI), 2021
> >
> > [4]Wei Fang, Zhaofei Yu, Yanqi Chen, Tiejun Huang, Timothée Masquelier, and Yonghong Tian. Deep residual learning in spiking neural networks. Advances in Neural Information Processing Systems (NeurIPS), 2021
> >
> > [5]Shikuang Deng, Yuhang Li, Shanghang Zhang, and Shi Gu. Temporal efficient training of spiking neural network via gradient re-weighting. In International Conference on Learning Representations(ICLR), 2021
> >
> > [6]Wulfram Gerstner, Werner M Kistler, Richard Naud, and Liam Paninski. Neuronal dynamics: From single neurons to networks and models of cognition. Cambridge University Press, 2014.

---

> > > ### Comment · Reviewer_Ljtf · 2022-08-07
> > > **Thanks for clarifying**
> > >
> > > Thank you for the detailed comments. The responses have addressed all my concerns and comments. After reading the other reviews, I still believe this paper presents a contribution worth of acceptance. Therefore, I would like to raise my score to 8.

---

> > > > ### Author Response · Authors · 2022-08-07
> > > > **Thank You**
> > > >
> > > > Thank you very much for increasing the score and for the detailed comments.

---

### Official Review · Reviewer_VwQw · 2022-07-10

**Rating:** 3
**Confidence:** 5
**Soundness:** 2 fair
**Presentation:** 2 fair
**Contribution:** 2 fair

**Summary:**

The authors proposed a batch normalization scheme considering the temporal domain, TEBN. The proposed TEBN requires all time-step information to estimate the desired expectation and variance. The authors provided theoretical analysis and validated the effectiveness of the proposed TEBN with classification tasks.

**Questions:**

Besides the questions, I listed above, the authors must need to describe the detailed architecture of the models used in their experiments. They cannot assume the audiences know them or let audiences play a guessing game.

**Limitations:**

The validation is not convincing at all from my perspective. Only LIF and classification tasks cannot show the generalization of the proposed method. Without revealing the performance of the tasks that rely on temporal information, I do not think the proposed scheme is meaningful to the community.

As such, I suggest the authors show that the proposed scheme works with time-sensitive tasks.

**Strengths And Weaknesses:**

Strengths:
+1. The authors showed good efforts in theoretical analysis, which is desired for the NeurIPS audience.

Weaknesses:

-1. Requiring all time steps information is not practical at all, especially for time-dependent tasks, e.g., tracking. For classification tasks, it may make sense. However, using SNN for classification tasks lacks justification as the tasks do not really need temporal information.

-2. The experimental setup is unclear, and validations are weak. For VGG, ResNet, and other architectures, which parts are based on SNNs? For the ResNet-19, did the authors only replace the ReLU with SNNs?
* The experimental results did not show the impacts of the following factor on the classification tasks.
     * mini-batch size
     * spiking threshold
     * higher time steps: The authors claimed that fewer time steps are enough for the tested classification tasks. However, how do the larger time steps impact the results?

-3. The experimental results are only based on LIF. What about other SNN neuron models, e.g., SRM, IF?

---

> ### Author Response · Authors · 2022-08-02
> **Response to Reviewer VwQw Part I/III**
>
> Thank you for your constructive comments and suggestions. We would like to address your concerns and answer your questions here.
>
> ***Comment 1.Requiring all time steps information is not practical at all, especially for time-dependent tasks, e.g., tracking. For classification tasks, it may make sense. However, using SNN for classification tasks lacks justification as the tasks do not really need temporal information.***
>
>
> Thanks for your comments. We would like to note that our proposed method currently aims to solve the problem of SNN surrogate training on deep architectures. The benchmark task of this field is the classification task. Our classification tasks are conducted not only on static image datasets, but also on event-based datasets, some of which are recognized as being naturally encoded temporal information [1].
>
> You have raised an interesting concern about how to require the information of all time steps for time-dependent tasks. There may be a misunderstanding about ''all timesteps''. In our paper, ''all timesteps'' means all the steps in a window instead of the entire data sequence. For the event-based classification task, we adjust the full length of the event data into a fixed window. The window has a pre-determined size called ''timestep''. For video-based tasks like tracking, the existence of window size means that one can feed multiple frames of a video into SNNs once a time.
>
>
> ***Comment 2.The experimental setup is unclear, and validations are weak. For VGG, ResNet, and other architectures, which parts are based on SNNs? For the ResNet-19, did the authors only replace the ReLU with SNNs?
> The experimental results did not show the impacts of the following factor on the classification tasks. mini-batch size, spiking threshold, higher time steps: The authors claimed that fewer time steps are enough for the tested classification tasks. However, how do the larger time steps impact the results?***
>
>
> We would like to declare our experimental setting: Yes, for VGG and ResNet architectures, we replace the ReLU activation with Leaky Integrate-and-fire neurons. The leaky factor of neurons can be manually set. Our TEBN directly trains SNN using surrogate gradients. Besides using the spiking neuron, the computational graph and gradients calculation is different from ANN due to the computational mechanism of the spiking neuron and additional temporal dimension. Overall, our setup of the model is identical to the related works [2][3][4] in the field of SNN training. Experiments details of the configurations and models are provided in Supplementary Material (Please refer to Sec. E of the Appendix).
>
> Thanks for the suggestions on the impacts of some experimental factors. We have added new experiments on the CIFAR-10 dataset to demonstrate the generalization performance of our model. Here we use the 7-layer spiking CNN with the structure 28C3-256C3-AP2-512C3-AP2-1024C3-512C3-1024FC-10FC. We set the batch size to 64, timestep to 4, and threshold to 1 as the default setting.
>
> We first experiment on the mini-batch size. The setting of batch size will potentially influence the statistics of our model. We set BatchSize=[4, 8, 16, 32, 64] and obtain the outcome Accuracy=[92.4%, 93.04%, 93.11%, 93.12%, 92.57%], as shown in Table R1. Our results demonstrate that though there is a significant change in batch size setting, our TEBN model still maintains good generalization capabilities.
>
> **Table R1: Comparison with different batch sizes on CIFAR-10 dataset.**
> |Batch |Accuracy(%)|
> |:-----:|:------------:|
> | 4 | 92.40 |
> |8 |   93.04 |
> |16 |   93.11 |
> |  32  |  93.12 |
> |64 |   92.57 |
>
>
> Besides, we test the effect of different thresholds. The firing threshold is a hyper-parameter of the spiking neuron, corresponding to a biological neuron characteristic. When the Threshold=[0.5, 1.0, 1.5], we obtain the Accuray=[92.22%, 92.57%, 92.99%] in experiments, as shown in Table R2. Our experimental results show that our TEBN generalizes well when changing the threshold value.
>
> **Table R2: Comparison with different thresholds on CIFAR-10 dataset.**
> |Threshold |Accuracy(%)|
> |:-----:|:------------:|
> | 0.5 |   92.22 |
> | 1.0 |   92.57 |
> | 1.5 |   92.99 |

---

> > ### Author Response · Authors · 2022-08-02
> > **Response to Reviewer VwQw Part II/III**
> >
> > As for the timestep of SNN, we present the results of 2,4 and 6 time-steps. This setting is the same as that in the SNN SOTA works [5]. Our results in Tables 2 and 3 imply that larger time-steps may increase the accuracy of SNN. Nevertheless, we would like to note that the field of SNN favors the research of generalization on fewer time-steps. A larger time-step means greater computation cost both in training and inference. The field of SNN in AI has come a long way to reducing the time-step required for inference and training: As suggested in our Related Work, ANN-to-SNN conversion ''demands many time-steps to approach the accuracy of pre-trained ANNs''. Compared to conversion methods,  backpropagation with surrogate gradients requires far fewer time-steps. Many works have been focused the performance on the extremely time-steps (less than 10)[4][5][6]. Therefore, it has always been the goal of SNN training to achieve higher performance with fewer time-steps.
> >
> > ***Comment 3.The experimental results are only based on LIF. What about other SNN neuron models, e.g., SRM, IF?***
> >
> > In terms of mimicking the brain network, we are aware of the fact that there are spectrums of models varying in their biological realistic and computational efficiency. The Integrate-and-Fire model (IF) is the simplest in biology and the most efficient for computation. There exist more complex models like the Leaky integrate-and-fire model (LIF), spiking response model(SRM), Hodgkin–Huxley model (HH), etc.
> >
> > LIF can be seen as a special case of SRM, and IF can be seen as a special case of LIF. LIF degenerates to the IF when membrane time constant $\tau$ used in our method is set to 1.0.
> > We have actually experimented on IF models. We showed the impacts of changing $\tau$ in Sec 6.4. We have chosen different $\tau$s to test the generalization, $\tau$=0.1, 0.25, 0.5, 0.75, 1.0. The accuracies of these $\tau$ settings are 93.39, 93.54, 93.5, 93.53, and 93.61. These results can prove that our model has the potential to be applied to other neuron models.
> >
> > We won't overclaim that our model can work for more complex neuron models without performing the related experiments. Indeed, how to build a computational-efficient model with complex neuron dynamics is a quite challenging problem for the whole machine learning and computational neuroscience groups. We are glad to investigate it further in future work.
> >
> > ***Comment 4. The authors must need to describe the detailed architecture of the models used in their experiments.***
> >
> > We would like to clarify that the detailed network architectures used in our experiments are provides in Sec. E2 of Supplementary Material.
> >
> > ***Comment 5.Only LIF and classification tasks cannot show the generalization of the proposed method. Without revealing the performance of the tasks that rely on temporal information, I do not think the proposed scheme is meaningful to the community.***
> >
> > Thanks for your comment. We would like to note that some event-based DVS datasets are recognized as being naturally encoded temporal information [1]. Besides, we added the experiment on Speech Commands dataset to validate the effectiveness of TEBN and report the results in Table R3. We apply the 4-layer CNN proposed in [7] and train the network for 50 epochs. We obtain accuracy of 95.47% with TEBN, 95.12% with BN, 92.33% with Layer Normalization (LN) and 94.60% without BN. The results illustrate that our TEBN can address issues related to sequential data better than some popular normalization methods.
> >
> > **Table R3: Comparison with other normalization on Speech Commands dataset.**
> >
> > |Model | Architecture |  Accuracy(%) |
> > |:-----:|:---:|:---------:|
> > | Without BN |4-layer CNN| 94.60 |
> > |LN | 4-layer CNN|  92.33 |
> > |BN |  4-layer CNN| 95.12 |
> > |  **TEBN**  |4-layer CNN|  **95.47** |

---

> > > ### Author Response · Authors · 2022-08-02
> > > **Response to Reviewer VwQw Part III/III**
> > >
> > > [1]Iyer, Laxmi R., Yansong Chua, and Haizhou Li. Is neuromorphic mnist neuromorphic? analyzing the discriminative power of neuromorphic datasets in the time domain. Frontiers in neuroscience 15, 2021
> > >
> > > [2]Hanle Zheng, Yujie Wu, Lei Deng, Yifan Hu, and Guoqi Li. Going deeper with directly trained larger spiking neural networks. In Proceedings of the AAAI Conference on Artificial Intelligence(AAAI), 2021
> > >
> > > [3]Youngeun Kim and Priyadarshini Panda. Revisiting batch normalization for training low-latency deep spiking neural networks from scratch. Frontiers in Neuroscience, 15,2021.
> > >
> > > [4]Wei Fang, Zhaofei Yu, Yanqi Chen, Tiejun Huang, Timothée Masquelier, and Yonghong Tian. Deep residual learning in spiking neural networks. Advances in Neural Information Processing Systems (NeurIPS), 2021
> > >
> > > [5]Shikuang Deng, Yuhang Li, Shanghang Zhang, and Shi Gu. Temporal efficient training of spiking neural network via gradient re-
> > > weighting. In International Conference on Learning Representations(ICLR), 2021
> > >
> > > [6]Yufei Guo, Xinyi Tong, Yuanpei Chen, Liwen Zhang, Xiaode Liu, Zhe Ma, and Xuhui Huang. RecDis-SNN: Rectifying Membrane Potential Distribution for Directly Training Spiking Neural Networks. In Proceedings of the IEEE/CVF Conference on Computer Vision and Pattern Recognition (CVPR), 2022
> > >
> > > [7]https://github.com/fangwei123456/spikingjelly/blob/master/spikingjelly/activation_based/examples/speechcommands.py

---

> > > > ### Comment · Area_Chair_oW97 · 2022-08-09
> > > > **Any feedback to the responses from authors?**
> > > >
> > > > Dear Reviewer VwQw,
> > > >
> > > > Thank you very much for your review.  The authors have provided detailed responses to your review.  Do they resolve your concerns, or do you still have anything you would like to clarify?  We will be finishing the rebuttal period soon, so please let us know your opinions after the responses from the authors.  Thank you!

---

> > > > > ### Comment · Reviewer_VwQw · 2022-08-09
> > > > > **Thanks for the responds**
> > > > >
> > > > > Dear Authors,
> > > > >
> > > > > Thanks so much for the efforts to try to address all my concerns!
> > > > >
> > > > > However, I am still not convinced.
> > > > >
> > > > > DVS (or Event-based camera) for classification lacks justification as DVS is sensitive to motion. Still, no motion is required for classification tasks.
> > > > >
> > > > > I agree with the authors that LIF is a special case of SRM. But, it does not mean working with LIF would work with SRM. I still suggest conducting SRM-based experiments to show the generalization of the proposed method.
> > > > >
> > > > > Threshold experiments do not make sense without showing the average potential. Based on my experiences, setting a large threshold would surely break the network as no signal would pass through.
> > > > >
> > > > > "all timesteps" means all the steps in a window. Sure! But, what's the window size? How does the window size or the number of windows impact the proposed approach?
> > > > >
> > > > > In summary, for my perspective, I do not think the work is in the shape to be accepted by NeurIPS. Therefore, I keep my original rate.

---

> > > > > > ### Author Response · Authors · 2022-08-09
> > > > > > **Further Response to Reviewer VwQw Part I/II**
> > > > > >
> > > > > > Dear Reviewer VwQw,
> > > > > >
> > > > > > Thank you for your constructive further comments and suggestions. We would like to address your concerns and answer your questions here.
> > > > > >
> > > > > > ***Comment 1. DVS (or Event-based camera) for classification lacks justification as DVS is sensitive to motion. Still, no motion is required for classification tasks.***
> > > > > >
> > > > > > We agree that no motion is required for classification tasks. We have added the experiment on the Speech Commands dataset to compare our method with other normalization methods. We apply the 4-layer CNN proposed in [1] and train the network for 50 epochs. As reported in Table R4, our method achieves better performance than other normalization methods, which implies that our method works for the time-dependent task. As you suggested, we test our method on the tracking task, and our experiments are still ongoing. Due to the limited rebuttal time, we would like to provide the results in the final version.
> > > > > >
> > > > > >
> > > > > > **Table R4: Comparison with other normalization methods on Speech Commands dataset.**
> > > > > > |Model | Architecture |  Accuracy(%) |
> > > > > > |:-----:|:---:|:---------:|
> > > > > > |LN | 4-layer CNN|  92.33 |
> > > > > > |BN |  4-layer CNN| 95.12 |
> > > > > > |BNTT[2]| 4-layer CNN| 94.23 |
> > > > > > |tdBN[3]| 4-layer CNN| 95.39 |
> > > > > > |  **TEBN**  |4-layer CNN|  **95.47** |
> > > > > >
> > > > > >
> > > > > > ***Comment 2. I agree with the authors that LIF is a special case of SRM. But, it does not mean working with LIF would work with SRM. I still suggest conducting SRM-based experiments to show the generalization of the proposed method.***
> > > > > >
> > > > > > Thanks for your suggestion. As the rebuttal time is due soon, we only added the experiment of SRM on the CIFAR-10 dataset to validate the generalization of TEBN. For the SRM model, we used standard double-exponential PSP kernels with a brief finite rise and exponential decay, of the form $\epsilon(t) = \frac{\tau_m}{\tau_m - \tau_s} (e^{-\frac{t}{\tau_m}} - e^{-\frac{t}{\tau_s}})$. Due to limited time, we only train the SRM model with 60 epochs. As shown in Table R5, the SRM-based network gets slighter better accuracy than the LIF-based network with the same structure, which implies that our method can be generalized to different neuron models.
> > > > > >
> > > > > >
> > > > > >
> > > > > > **Table R5: Performance on CIFAR-10 dataset with SRM model.**
> > > > > > |Architecture | Neuron |Epoch|  Accuracy(%) |
> > > > > > |:-----:|:---:|:-----:|:-----:|
> > > > > > | 7-layer CNN |LIF($\tau$=0.25)| 60/100 |90.30/92.57|
> > > > > > |7-layer CNN | SRM（${\tau_s}$=2,${\tau_m}$=4）| 60  |90.32
> > > > > >
> > > > > > ***Comment 3. Threshold experiments do not make sense without showing the average potential. Based on my experiences, setting a large threshold would surely break the network as no signal would pass through.***
> > > > > >
> > > > > > Thanks for your suggestion, we have calculated the average potential at each layer and reported the results in Table R6. We can find that a larger threshold makes a larger average potential. As the accuracies change very little for the thresholds 0.5, 1, and 1.5, our method obtains good scalability when threshold changing. Furthermore,  we have added new experiments on the CIFAR-10 dataset with much larger thresholds and reported the results in Table R7. Due to limited time for rebuttal, we only train the networks with 36 epochs. We find that a larger threshold makes the training slower to converge. Besides, the training fails when the threshold is larger than 5.5.  These results are consistent with your suggestion that "a large threshold would surely break the network as no signal would pass through."
> > > > > >
> > > > > > **Table R6: Average potential at each layer with different thresholds.**
> > > > > > |Threshold |Layer 1 | Layer 2 | Layer 3| Layer 4 |Layer 5 | Epoch|Accuracy(%)|
> > > > > > |:-----:|:----:|:---:|:----:|:--:|:----:|:----:|:---:|
> > > > > > | 0.5 |0.06|   -0.29    |  -0.34  |  -0.36 | -0.07|100|92.22|
> > > > > > | 1.0 | 0.30|   -0.20   |  -0.14  |  -0.13 | 0.20|100|92.57|
> > > > > > | 1.5 | 0.48|   0.01   |  0.05  |  0.03 | 0.38|100|92.99|
> > > > > >
> > > > > > **Table R7: Comparison with different thresholds on CIFAR-10 dataset.**
> > > > > > |Architecture|Threshold |Epoch|Accuracy(%)|
> > > > > > |:-----:|:------:|:---:|:---:|
> > > > > > |7-layer CNN |2.0 |    36 |90.21|
> > > > > > |  7-layer CNN| 2.5 |    36 |88.82|
> > > > > > |  7-layer CNN| 3.0  |  36 |87.85|
> > > > > > | 7-layer CNN |4.0 |  36  | 86.33|
> > > > > > |  7-layer CNN |5.0 |    36 | 83.08|
> > > > > >
> > > > > > ***Comment 4. "all timesteps" means all the steps in a window. Sure! But, what's the window size? How does the window size or the number of windows impact the proposed approach?***
> > > > > >
> > > > > > We would like to note that the setting of timesteps means that we have chosen fixed window size. The window size has an influence on the final classification accuracy as it can increase the precision of sampling. Data of all timesteps in our classification task has first been squeezed into a window. So far, we have not researched the effect of multiple windows. As for the window size, we present the results of 2,4, and 6 timesteps for our proposed approach. Our results in Tables 2 and 3 of the manuscript imply that larger timesteps may increase the accuracy of SNNs.

---

> > > > > > > ### Author Response · Authors · 2022-08-09
> > > > > > > **Further Response to Reviewer VwQw Part II/II**
> > > > > > >
> > > > > > > [1]https://github.com/fangwei123456/spikingjelly/blob/master/spikingjelly/activation_based/examples/speechcommands.py
> > > > > > >
> > > > > > > [2]Youngeun Kim and Priyadarshini Panda. Revisiting batch normalization for training low-latency deep spiking neural networks from scratch. Frontiers in Neuroscience, 15:773954–773954, 2021.
> > > > > > >
> > > > > > > [3]Hanle Zheng, Yujie Wu, Lei Deng, Yifan Hu, and Guoqi Li. Going deeper with directly trained larger spiking neural networks. In Proceedings of the AAAI Conference on Artificial Intelligence(AAAI), 2021

---

> ### Author Response · Authors · 2022-08-07
> **Thanks for your valuable time and we would like to know if there is any further concern**
>
> Dear Reviewer VwQw,
>
> Thank you for the thorough feedbacks and constructive suggestions. Since it is approaching the end of author-reviewer discussion period, we would like to kindly ask if our previous response clarifies your concerns and if there are any further comments, and we are glad to cooperate and answer to facilitate the review process. Thanks a lot for your time!

---

### Official Review · Reviewer_RAfA · 2022-07-10

**Rating:** 6
**Confidence:** 4
**Soundness:** 3 good
**Presentation:** 3 good
**Contribution:** 2 fair

**Summary:**

The manuscript presents a technique (TEBN) for batch normalization in SNNs that takes into account the temporal dimension of the spiking input to rescale and recenter the input distribution differently at each input timestep, even though it still estimates the mean and variance of the whole batch, throughout all timesteps. The paper's theory shows that this smoothens the loss landscape. Experiments show that TEBN improves accuracy, robustness, and latency compared to other batchnorm methods and compare to the state of the art in the SNN literature, in image recognition of common benchmark datasets, including recordings of static images through neuromorphic vision sensors.

**Questions:**

Could the authors please respond to the above points, and ideally address them with experiments?
Are the accuracies of other methods reported in the tables results from the authors own experiments, or from the literature? Please clarify in the paper.

**Limitations:**

The authors have not discussed any limitations of the work.

**Strengths And Weaknesses:**

The paper targets a true problem in the field of SNNs, and appears to provide a truly improved solution for certain important cases. In addition, it provides theoretical analysis, and experimental insights deeper than merely end accuracies, showing the impact of the method on input distributions and on robustness to hyperparameters.

On the other hand, the paper:
- does not address issues related to truly sequential data. SNNs are considered as most suitable for temporal tasks, therefore a complete evaluation of the method should involve such tasks. An example is keyword spotting on the Speech Commands dataset. Currently, all tested datasets in the paper only include static images, even if in some cases the images are recorded through a DVS.
- does not compare to layer normalization, a technique known as suitable for recurrent networks and temporal datasets, even though the authors' input includes a temporal dimension, and even though LIF SNNs are recurrent networks.
- does not explain how its results compare to the true state of the art, i.e. beyond SNNs. For example, non-spiking networks can also reduce latency and computation while maintaining their higher accuracy than SNNs (Jeffares et al., ICLR 2022 https://openreview.net/forum?id=iMH1e5k7n3L). But more generally, it would be good to put the results back in the context of the broader ML field, and discuss the differences e.g. in accuracy, latency, or computational efficiency.
- does not mention the method's strong similarity to short-term plasticity (STP), which is a long-known property of biological SNNs for temporal filtering of inputs (see e.g. Fortune & Rose, 2001 https://roselab.biology.utah.edu/publications/trends2001.pdf; Rosenbaum et al., 2012 https://journals.plos.org/ploscompbiol/article?id=10.1371/journal.pcbi.1002557). Temporal filtering is what the authors' TEBN also applies. Of course, STP applies a fixed, e.g. exponential kernel, whereas TEBN learns the shape of the temporal kernel. On the other hand, short-term plasticity is also learnable (Garcia-Rodriguez et al., ICML 2022 https://arxiv.org/abs/2206.14048), which then makes STP similar to what the authors here achieve, but in several aspects with more flexibility than what the authors here propose. Moreover, STP has been shown to be a powerful property of SNNs, such that SNNs with STP can even outperform LSTMs and CNNs in accuracy (Moraitis et al., 2021 https://arxiv.org/abs/2009.06808). Even though the authors do not have to perform experimental comparisons of their method against STP, STP's relevance through these works must be mentioned, and discussed as a potential alternative for future work, that is also more biologically-realistic.

---

> ### Author Response · Authors · 2022-08-02
> **Response to Reviewer RAfA Part I/II**
>
> Thank you for your constructive  comments, suggestions and appreciation of our work. We would like to address your concerns and answer your questions here.
>
>
>
> ***Comment 1. does not address issues related to truly sequential data. SNNs are considered as most suitable for temporal tasks, therefore a complete evaluation of the method should involve such tasks. An example is keyword spotting on the Speech Commands dataset. Currently, all tested datasets in the paper only include static images, even if in some cases the images are recorded through a DVS.***
>
> Thanks for your suggestion. We have added the experiment on Speech Commands dataset to validate the effectiveness of TEBN and report the results in Table R1. We apply the 4-layer CNN proposed in [1] and train the network for 50 epochs. We obtain accuracy of 95.47% with TEBN, 95.12% with BN, 92.33% with Layer Normalization (LN) and 94.60% without BN. The results illustrate that our TEBN can address issues related to sequential data better than some popular normalization methods.
>
>
> **Table R1: Comparison with other normalization on Speech Commands dataset.**
>
> |Model | Architecture |  Accuracy(%) |
> |:-----:|:---:|:---------:|
> | Without BN |4-layer CNN| 94.60 |
> |LN | 4-layer CNN|  92.33 |
> |BN |  4-layer CNN| 95.12 |
> |  **TEBN**  |4-layer CNN|  **95.47** |
>
>
>
>
> ***Comment 2. does not compare to layer normalization, a technique known as suitable for recurrent networks and temporal datasets.***
>
> Thanks for your valuable suggestions. BN takes the same feature of different samples, while LN takes the different features of the same sample. To compare with LN, we have conducted multiple sets of experiments. In addition to the experiment on Speech Commands Dataset (Table R1), we also compare the performance of LN and the proposed TEBN on CIFAR-10 and DVS-CIFAR10 datasets (shown in Tables R2 and R3). Here we train the 7-layer CNN on CIFAR10 and the 6-layer CNN on DVS-CIFAR10 using LN, with the same networks as Table 3 in the manuscript. From the observation of our experiment, we find that our TEBN outperforms LN on CIFAR-10 and DVS-CIFAR10. The performance of TEBN is 92.65%  (v.s. 83.79% of LN) on CIFAR10 and 80.00% (v.s. 62.90% of LN) on DVS-CIFAR10. Our results are consistent with the conclusion of [2] that BN generally performs better than LN in CNN.
>
>
> **Table R2: Comparison with LN on CIFAR-10 dataset.**
> |Model | Architecture | Time-steps| Accuracy(%) |
> |:-----:|:------------:|:----:|:-------:|
> LN     |   7-layer CNN | 4| 83.79  |
> **TEBN**   |   7-layer CNN | 4| **92.65**  |
>
> **Table R3: Comparison with LN on DVS-CIFAR10 dataset.**
> |Model | Architecture | Time-steps| Accuracy(%) |
> |:-----:|:------------:|:----:|:-------:|
> LN     |   6-layer CNN | 10| 62.90  |
> **TEBN**   |   6-layer CNN | 10| **80.00**  |
>
>
> ***Comment 3. does not explain how its results compare to the true state of the art, i.e. beyond SNNs. For example, non-spiking networks can also reduce latency and computation while maintaining their higher accuracy than SNNs (Jeffares et al., ICLR 2022). But more generally, it would be good to put the results back in the context of the broader ML field, and discuss the differences e.g. in accuracy, latency, or computational efficiency.***
>
> Thanks for your constructive suggestion. Back in the context of the broader ML field, we would like to include the result of non-spiking ANN SOTA in the comparison. Using the similar network architectures, the ANN accuracies are 95.55%[3] on CIFAR-10 and 78.49%[4] on CIFAR-100 (Tables R4 and R5).
>
> We would like to note that computation efficiency is the benefit of SNN. Our TEBN does not directly cope with the problem of computational efficiency. In the context of SNN, binary activation eliminates the multiplication through adding operation, which theoretically leads to computational efficiency if the hardware supports the deployment, e.g. Loihi chips from Intel, where 0 activations (no spike) will not be involved in the computation. Detailed discussions of computational efficiency between non-spiking network and SNN can be found in [5].
>
> **Table R4: Comparison with ANN on CIFAR-10 dataset.**
>
> |Model | Architecture | Time-steps| Accuracy(%) |
> |:-----:|:------------:|:----:|:-------:|
> ANN[3]    |    ResNet-18  |     1     |  95.55  |
> **TEBN**   |    ResNet-19  |     2     |  95.45  |
>
> **Table R5: Comparison with ANN on CIFAR-100 dataset.**
> |Model | Architecture | Time-steps| Accuracy(%) |
> |:-----:|:------------:|:----:|:-------:|
> ANN[4]    |    PreAct-ResNet-18  |    1     |  78.49  |
> **TEBN**   |    ResNet-19  |    2     |  78.07  |

---

> > ### Author Response · Authors · 2022-08-02
> > **Response to Reviewer RAfA Part II/II**
> >
> > ***Comment 4. does not mention the method's strong similarity to short-term plasticity (STP), which is a long-known property of biological SNNs for temporal filtering of inputs (see e.g. Fortune & Rose, 2001 ; Rosenbaum et al., 2012 ). Temporal filtering is what the authors' TEBN also applies. Of course, STP applies a fixed, e.g. exponential kernel, whereas TEBN learns the shape of the temporal kernel. On the other hand, short-term plasticity is also learnable (Garcia-Rodriguez et al., ICML 2022 ), which then makes STP similar to what the authors here achieve, but in several aspects with more flexibility than what the authors here propose. Moreover, STP has been shown to be a powerful property of SNNs, such that SNNs with STP can even outperform LSTMs and CNNs in accuracy (Moraitis et al., 2021 ). Even though the authors do not have to perform experimental comparisons of their method against STP, STP's relevance through these works must be mentioned, and discussed as a potential alternative for future work, that is also more biologically-realistic.***
> >
> > Thanks for your suggestion. STP is indeed relevant and we have now cited and discussed in the revised paper. In the conclusion section, we added "Besides, recent works have shown that short-term plasticity (STP) \cite{fortune2001short,rosenbaum2012short,tsodyks1997neural} can be incorporated into ANNs to enhance efficiency and computational power \cite{moraitis2020optimality,rodriguez2022short}. As STP performs a function of temporal filtering similar to TEBN, how to use the biologically-realistic filter-STP in SNNs is the future direction.
> >
> >
> > ***Comment 5. Are the accuracies of other methods reported in the tables results from the authors own experiments, or from the literature? Please clarify in the paper.***
> >
> > All the accuracies of other methods reported in the tables are results from the mentioned literature. We have now clarified the source in the revised manuscript.
> >
> >
> >
> > [1]https://github.com/fangwei123456/spikingjelly/blob/master/spikingjelly/activation_based/examples/speechcommands.py
> >
> > [2]Ba, Jimmy Lei, Jamie Ryan Kiros, and Geoffrey E. Hinton. Layer normalization. arXiv preprint arXiv:1607.06450, 2016.
> >
> > [3]Moreau, Thomas, et al. Benchopt: Reproducible, efficient and collaborative optimization benchmarks. arXiv preprint arXiv:2206.13424, 2022.
> >
> > [4]Zheng, Yaowei, Richong Zhang, and Yongyi Mao. "Regularizing neural networks via adversarial model perturbation." In Proceedings of the IEEE/CVF Conference on Computer Vision and Pattern Recognition(CVPR), 2021.
> >
> > [5]Parker, Luke, Frances Chance, and Suma Cardwell. "Benchmarking a Bio-inspired SNN on a Neuromorphic System." In Neuro-Inspired Computational Elements Conference, 2022.

---

> > > ### Comment · Reviewer_RAfA · 2022-08-07
> > > **Update**
> > >
> > > Taking into account the authors' responses to other reviewers and me, I am updating the rating. I found the rebuttal convincing.

---

> > > > ### Author Response · Authors · 2022-08-07
> > > > **Thank You**
> > > >
> > > > Thank you very much for the thorough review and for increasing the score.

---

### Official Review · Reviewer_gvBu · 2022-07-12

**Rating:** 4
**Confidence:** 5
**Soundness:** 3 good
**Presentation:** 3 good
**Contribution:** 2 fair

**Summary:**

The paper proposes a temporal BN method to train SNNs with high accuracy,

**Questions:**

My main concern is teh technical novelty of this paper with respect to previous works. The authors have performed a holistic comparison. But, can they comment on how their work is technically novel? At this time, in my opinion, the paper reads to be more incremental effort.

**Limitations:**

Please see the above comments on weaknesses.

**Strengths And Weaknesses:**

+ The methos is simple and effective as suggested by author's results.
-The main comment I ahve is on the novelty of this work. The authors have cited many relevant works such as BNTT, TDBN etc. I think the paper's results are very similar to that of BNTT. Except for the fact that the accuracy is better in the datasets that authros have exeprimented with, I don't thinkt here is a lot of technical contribution or novelty.
-Further, authors of BNTT and TDBN experimented on larger datasets like Tiny Imagenet. Can the authors commnet on the scalability of their approach?
-BNTT and TDBN like methods were adapted for learning in diverse scenarios, liek Federated learning [1], Segmentation [2], large scale DVS datasets, NAS optimization[3] among others. Maybe the authors can include a commentary or discussion on how their approach can be applied to more diverse learning scenarios.

[1] Venkatesha, Yeshwanth, et al. "Federated learning with spiking neural networks." IEEE Transactions on Signal Processing 69 (2021): 6183-6194.
[2] Kim, Youngeun, et al. "Beyond classification: directly training spiking neural networks for semantic segmentation." arXiv preprint arXiv:2110.07742 (2021).
[3]Kim, Youngeun, et al. "Neural architecture search for spiking neural networks." arXiv preprint arXiv:2201.10355 (2022).

---

> ### Author Response · Authors · 2022-08-02
> **Response to Reviewer gvBu Part I/II**
>
>
>
>
> Thank you for your detailed and insightful comments. We are encouraged that you find our method effective. We would like to address your concerns and answer your questions here.
>
> ***Comment 1. The main comment I have is on the novelty of this work. The authors have cited many relevant works such as BNTT, TDBN etc. I think the paper's results are very similar to that of BNTT.***
>
> We would like to clarify that our work is different from BNTT, tdBN, etc. As illustrated in Sec. 2.2, our work is inspiration from these related works and takes advantage of them. To be specific, BNTT can utilize separate sets of BN parameters on different time-steps to mitigate the temporal shift of distributions. While in tdBN, the utilization of shared parameters may neglect the negative impact brought by the unusual temporal distributions. The most significant difference of this work is: our TEBN model can model the temporal shift of distributions without including T times volumes of BN parameters (Eq.10-13) and take advantage of the overall distribution (Theorem 1&2). We believe our work will be appealing for the implementation of energy-efficient SNNs (fewer parameters) and the theoretical generalization capability of SNNs (better experimental results).
>
> ***Comment 2. The authors of BNTT and TDBN experimented on larger datasets like Tiny Imagenet. Can the authors comment on the scalability of their approach?***
>
> Thanks for your suggestion. We have performed new experiments on the ImageNet dataset to demonstrate the scalability of our method. We first compare the performance of the proposed TEBN and tdBN [1]. As reported in Table R1, our method achieves better performance (64.29\% v.s. 63.72\%) and fewer time-steps  (4 v.s. 6) than tdBN [1]. Then we compare our method with other state-of-the-art learning methods [2,3] for SNNs. One can find that our method outperforms vanilla SEW [2] and TET [3] when the architecture and time-steps are the same. All these results demonstrate that the proposed method can scale to larger datasets. We have added these results in Tables 2 and 3 of the revised paper.
>
>
> **Table R1: Comparison with other normalization method and the SOTA training methods on ImageNet dataset.**
> |Model |Methods | Architecture | Time-steps| Accuracy(%) |
> |:-----:|:------:|:------:|:----:|:-------:|
> | tdBN[1] |Surrogate Gradient|   ResNet-34    |  6  |  63.72 |
> | **TEBN** |Surrogate Gradient|   ResNet-34    |  4  |  **64.29** |
> | SEW[2] |Surrogate Gradient|   SEW ResNet-34    |  4  |  67.04 |
> | TET[3] |Surrogate Gradient|   SEW ResNet-34    |  4  |  68.00 |
> | **TEBN** |Surrogate Gradient|  SEW ResNet-34 |  4  | **68.28**  |

---

> > ### Author Response · Authors · 2022-08-02
> > **Response to Reviewer gvBu Part II/II**
> >
> > ***Comment 3. BNTT and TDBN like methods were adapted for learning in diverse scenarios, liek Federated learning, Segmentation, large scale DVS datasets, NAS optimization among others. Maybe the authors can include a commentary or discussion on how their approach can be applied to more diverse learning scenarios***
> >
> > You have raised an interesting concern. We have added the following references [4-6] in the revised paper to illustrate BNTT like methods can be used for learning in diverse scenarios.
> >
> >
> > Besides, we believe that our method also can be applied to more diverse learning scenarios. Due to limited time for rebuttal, here we add the experiment of federated learning [7] to validate the generalization of the proposed TEBN. We compare our method with BNTT. For a fair comparison, we use the same VGG-9 structure and only replace BNTT by TEBN. The results are shown in Table R2, we obtained final testing accuracy of 85.81%(v.s. 76.44% of BNTT) with 10 clients in total and 2 participating clients. We have added the comparison in the revised paper (please refer to the supplementary).
> >
> >
> >
> > **Table R2: Comparison with BNTT on the task of federated learning.**
> > |Model |Methods | Architecture | Time-steps| Accuracy(%) |
> > |:-----:|:-------:|:-----:|:----:|:-------:|
> > | BNTT[8] |Surrogate Gradient|   VGG-9    |  20  |  76.44 |
> > | **TEBN** |Surrogate Gradient|   VGG-9    |  4  |  **85.81** |
> >
> >
> >
> >
> >
> > [1]Hanle Zheng, Yujie Wu, Lei Deng, Yifan Hu, and Guoqi Li. Going deeper with directly trained larger spiking neural networks. In Proceedings of the AAAI Conference on Artificial Intelligence(AAAI), 2021
> >
> > [2]Wei Fang, Zhaofei Yu, Yanqi Chen, Tiejun Huang, Timothée Masquelier, and Yonghong Tian. Deep residual learning in spiking neural networks. Advances in Neural Information Processing Systems (NeurIPS), 2021
> >
> > [3]Shikuang Deng, Yuhang Li, Shanghang Zhang, and Shi Gu. Temporal efficient training of spiking neural network via gradient re-weighting. In International Conference on Learning Representations(ICLR), 2021
> >
> > [4] Venkatesha, Yeshwanth, et al. "Federated learning with spiking neural networks." IEEE Transactions on Signal Processing, 69: 6183-6194, 2021.
> >
> > [5] Kim, Youngeun, et al. "Beyond classification: directly training spiking neural networks for semantic segmentation." arXiv preprint arXiv:2110.07742, 2021.
> >
> > [6]Kim, Youngeun, et al. "Neural architecture search for spiking neural networks." European Conference on Computer Vision (ECCV), 2022.
> >
> > [7]https://github.com/Intelligent-Computing-Lab-Yale/FedSNN
> >
> > [8]Youngeun Kim and Priyadarshini Panda. Revisiting batch normalization for training low-latency deep spiking neural networks from scratch. Frontiers in Neuroscience, 15:773954–773954, 2021.

---

> > > ### Comment · Area_Chair_oW97 · 2022-08-09
> > > **Any feedback to the responses from authors?**
> > >
> > > Dear Reviewer gvBu,
> > >
> > > Thank you very much for your review.  The authors have provided detailed responses to your review.  Do they resolve your concerns, or do you still have anything you would like to clarify?  We will be finishing the rebuttal period soon, so please let us know your opinions after the responses from the authors.  Thank you!

---

> ### Author Response · Authors · 2022-08-07
> **Thank you for the time and hope that our response help for your assessment of our work**
>
> Dear Reviewer gvBu,
>
> We thank you for your detailed initial comments. We really hope to know whether our previous response has addressed your questions and concerns properly. As the discussion period will end soon, please let us know if you have any further questions that we would write a follow-up response. Thank you very much!

---

### Author Response · Authors · 2022-08-06
**Look Forward to Feedbacks**

Thank you for the detailed feedbacks and constructive suggestions. It has been several days since we submitted our response, but we have not received your reply. We spend a lot of effort in rebuttal, and we really hope that you can check whether our responses have addressed your concerns. Please let us know if you have any further comments, and we are glad to write a follow-up response. Thank you very much!

---

### Public Comment · ~Chaoteng_Duan1 · 2023-05-23
**Codes are available at Github**

Codes are available at https://github.com/ChaotengDuan/TEBN

---

### Meta-Review · Area_Chair_oW97 · 2022-08-25

**Recommendation:** Accept
**Confidence:** Less certain

**Metareview:**

The paper proposes a method of batch normalization that takes into account the temporal dimension (TEBN) and empirically shows that TEBN can significantly improve the accuracy of spiking neural networks (SNNs).  Theoretical analysis also provides new insights into how SNNs should be trained to improve accuracy (particularly in the face of temporal variation of internal covariate shift).

This paper had received conflicting evaluations from Reject to Strong Accept, and the reviewers did not reach a consensus even after fairly intense discussion.  The disagreement appears to come from what ones expect from SNNs: accuracy, robustness, latency, sparsity, biological plausibility, etc.  While it is well empirically supported (particularly with the additional experiments during rebuttal) that TEBN increases accuracy, TEBN certainly looses biological plausibility, and the operations such as variance computation needed in TEBN might not be desirable for some applications of SNNs.  Also, since much of the experiments are added during rebuttal, there is a criticism for the lack of consistency in experimental design, which also leads to mixed evaluations regarding the benefit of the proposed approach.

Overall, despite several weaknesses and uncertainties, the high accuracy certainly matters to some of the users and researchers of SNNs, and the paper clearly excels in this regard.  Hence, I recommend an acceptance.

**Award:**

No

---

### Decision · Program_Chairs · 2022-09-14

Accept